# A highly potent human neutralizing antibody prevents vertical transmission of Rift Valley fever virus in a rat model

Cynthia M. McMillen[1,2], Nathaniel S. Chapman[3], Ryan M. Hoehl[1], Lauren B. Skvarca[4], Madeline M. Schwarz[1,2], Laura S. Handal[5], James E. Crowe Jr. [3,5,6] ✉ & Amy L. Hartman [1,2] ✉

Rift Valley fever virus (RVFV) is an emerging mosquito-transmitted virus that circulates in livestock and humans in Africa and the Middle East. Outbreaks lead to high rates of miscarriages in domesticated livestock. Women are also at risk of vertical virus transmission and late-term miscarriages. MAb RVFV-268 is a highly potent recombinant neutralizing human monoclonal antibody that targets RVFV. Here we show that mAb RVFV-268 reduces viral replication in rat placenta explant cultures and prevents vertical transmission in a rat model of congenital RVF. Passive transfer of mAb RVFV-268 from mother to fetus occurs as early as 6 h after administration and persists through 24 h. Administering mAb RVFV-268 2 h prior to RVFV challenge or 24 h post-challenge protects the dams and offspring from RVFV infection. These findings support mAb RVFV-268 as a pre- and post-infection treatment to subvert RVFV infection and vertical transmission, thus protecting the mother and offspring.

Rift Valley fever (RVF) is an emerging zoonotic disease afflicting humans and livestock (primarily sheep, goats, cattle, camels) throughout Africa and regions of the Arabian Peninsula[1–3]. Rift Valley fever virus (RVFV; family *Phenuiviridae*, genus phlebovirus) was first identified in the Rift Valley of Kenya in 1932 after reports of unusually high rates of death in newborn lambs and abortions from ewes without apparent signs of disease[4]. High rates of concurrent fetal death in livestock, also known as "abortion storms", continue to affect sub-Saharan Africa[5]. Within a given outbreak, up to 95% of infected pregnant sheep can lose their young, often delivering deceased and deformed offspring[3,6]. Adult livestock typically develop mild signs of RVF disease, whereas juvenile animals are more susceptible to severe hepatic disease. The loss of almost an entire generation of livestock has led to significant socioeconomical strains in affected regions[7,8].

Transmission of RVFV to humans by mosquito bite or through contact with contaminated tissues of infected livestock often results in mild flu-like symptoms. Although less than 5% of infected individuals develop severe disease in the form of encephalitis, hemorrhagic fever, or ocular disease[9], approximately 50% of severe cases result in death[10]. Vertical transmission was exclusively reported in livestock until the mid-2000s when two isolated cases of human vertical transmission were documented between mothers and their late-gestation fetuses[11,12]. Both newborns were born with clinical signs of RVF hepatitis that consequently led to the death of at least one of the infants. More recent epidemiological data report that pregnant women with confirmed RVFV infection during pregnancy are more likely to have late-term fetal loss (odds ratio 7.4), suggesting miscarriages associated with RVF in humans may be underreported[13].

Live-attenuated RVFV vaccines are used to control outbreaks in livestock in endemic areas. However, teratogenic events can occur in pregnant animals with certain live-attenuated vaccines, thus restricting the use of these vaccines to non-pregnant animals[14,15]. Multiple

[1]University of Pittsburgh, Center for Vaccine Research, Pittsburgh, PA, USA. [2]University of Pittsburgh, Department of Infectious Diseases and Microbiology, School of Public Health, Pittsburgh, PA, USA. [3]Vanderbilt University Medical Center, Department of Pathology, Microbiology and Immunology, Nashville, TN, USA. [4]University of Pittsburgh Medical Center, Magee-Womens Hospital, Department of Pathology, Pittsburgh, PA, USA. [5]Vanderbilt University Medical Center, Vanderbilt Vaccine Center, Nashville, TN, USA. [6]Vanderbilt University Medical Center, Department of Pediatrics, Nashville, TN, USA. ✉ e-mail: james.crowe@vumc.org; hartman2@pitt.edu

institutions, including the World Health Organization, have listed RVFV as a priority pathogen with epidemic or pandemic potential due to a lack of vaccines and therapeutic interventions for people, thus highlighting a considerable need for more research in this area[16]. The prophylactic and therapeutic clinical opportunities presented by human monoclonal antibodies are becoming apparent. The use of human monoclonal antibodies to prevent disease from human immunodeficiency virus[17], SARS-CoV-2[18,19], respiratory syncytial virus[20], and *Plasmodium falciparum*[21] gives precedence that human monoclonal antibodies may help prevent RVFV infection and disease. Furthermore, the recent observation that individuals naturally infected with RVFV in Kenya make neutralizing antibodies targeting the glycoprotein Gn[22] and the successful isolation of antibodies targeting Gn hold promise that such antibodies can be developed into prevention measures[23,24]. One such mAb, RVFV-268, displayed ultra-potent neutralization against RVFV and mapped to the surface exposed residues of domain A on the Gn surface[23].

Here we show that prophylactic delivery of mAb RVFV-268 2 hours (h) prior to infection and up to 24 h post-infection protects mice from lethal subcutaneous challenge with RVFV ZH501[23]. RVFV causes a broad range of diseases in livestock and humans; therefore, a successful therapeutic should manage the broad-tropisms and pathologies associated with RVF. Here, we tested the ability of mAb RVFV-268 to prevent vertical transmission in a model of congenital RVF in Sprague-Dawley (SD) rats[25,26]. We found that mAb RVFV-268 rapidly reaches placental and fetal tissue after intraperitoneal injection, and that it prevents maternal and fetal infection in a dose-dependent manner when administered prior to challenge with RVFV. It also prevents vertical transmission when given up to 24 h after RVFV challenge, demonstrating its potential usefulness for either prophylaxis and/or therapy.

## Results

### RVFV-268 blocks RVFV infection of ex vivo placental tissue

RVFV displays extreme tropism for placental tissue in SD rats, as demonstrated by elevated virus titers in vivo in placenta within a few days of infection and a high rate of vertical transmission to fetuses[25]. In vitro, mAb RVFV-268 (IgG1) has shown ultra-potent virus neutralizing activity against several strains of RVFV, with a half maximal inhibitory concentration ($IC_{50}$) of approximately 0.2 ng/mL[23]. To determine whether mAb RVFV-268 can protect placental tissue from RVFV infection, we used ex vivo rat placenta slice cultures. Organ slice cultures have been successfully employed as a surrogate for in vivo experiments for various applications and organs[27–29], including the study of infectious diseases[30,31]. Rodent placenta slice cultures have been used to study local inflammatory responses[32] and other diseases[33]. For our studies, pregnant SD rats were euthanized at embryonic day 14 (E14) of a 22 day gestational period, corresponding to the gestational age at which we inoculate the rats with RVFV. Placentas were collected and immediately embedded in agarose cubes to provide support as each whole placenta was cut into 300 μm slices using a vibratome. Slices of similar sizes were cultured for antibody studies (Fig. 1A).

To determine if mAb RVFV-268 can prevent or reduce viral infection when added to placental tissue at the time of infection, 10 ng of mAb RVFV-268 or a control antibody that targets dengue virus, mAb DENV-2D22, was mixed with $1 \times 10^5$ pfu of RVFV ZH501 and added to the slice cultures during a 1 h adsorption period. The antibody-virus mixture was removed from the culture and 10 ng of the corresponding antibody was replenished in the growth medium. Supernatant was collected at 0, 24, 36, and 48 h post-inoculation (hpi) to generate viral growth curves (Fig. 1B, top). When added at the time of infection, mAb RVFV-268 significantly reduced infectious viral titers (Fig. 1B, left) and viral RNA (vRNA) (Fig. 1B, right) in the culture supernatant at 24, 36, and 48 h after inoculation. Titers of infectious virus and vRNA in the

mAb DENV-2D22 cultures were comparable to the no mAb control group. Similar reductions were observed in cultures containing 1-3-$\log_{10}$ higher concentration of mAb (Supp Figure 1A).

To determine if mAb RVFV-268 can prevent or reduce viral infection when added 1 h after infection in a therapeutic format, slices of similar size were generated then inoculated with $1 \times 10^5$ pfu RVFV followed by treatment with 10 ng of mAbs RVFV-268 or control DENV-2D22 after the virus adsorption period. MAb was kept in the growth medium for the duration of the experiment. Supernatant was collected at the same time points to generate viral growth curves (Fig. 1C, top). Addition of mAb RVFV-268 after virus inoculation marginally reduced infectious virus (Fig. 1C, right) and vRNA (Fig. 1C, left) throughout the virus growth period. Generally, reductions were observed at earlier time points in infection and did not persist through 48 h post inoculation. Infectious virus titers and vRNA were similar in the mAb DENV-2D22 and no mAb control groups. Similar results were obtained in cultures containing 100 ng, 1 μg, or 10 μg of mAb (Supp Figure 1B). The ex vivo placental cultures demonstrate that mAb RVFV-268 can reduce RVFV infection of placental tissue justifying the need to test its efficacy in an in vivo vertical virus transmission model.

### MAb RVFV-268 is rapidly transferred to placenta and fetuses in pregnant rats

To understand if mAb RVFV-268 crosses the placenta and reaches the fetuses, we injected 10 mg/kg of mAb RVFV-268 or control mAb DENV-2D22 intraperitoneally (i.p.) at E14 (n = 3 each), and then one dam from each antibody treatment group was euthanized at 6, 24 or 48 h after antibody delivery. Upon euthanasia, maternal (serum, liver, and spleen) and fetal (amniotic fluid, placenta, and viscera) samples were collected for measurement of mAb RVFV-268 by ELISA (Fig. 2A). High levels of mAb RVFV-268 were detected in all maternal and fetal samples by 6 h post-delivery of the antibody (Fig. 2B). MAb levels were lower at 24 h and were mostly undetectable by 48 h. Overall, antibody levels were highest in the maternal serum, liver, spleen, and the amniotic fluid, while lower antibody titers were found in the placenta and the viscera of the fetus. Samples from mAb DENV-2D22 dams were below the limit of detection for binding to RVFV Gn protein for all time points. The presence of mAb RVFV-268 in the placenta and fetal viscera indicates that this human antibody can passively transfer from the mother to the fetus *in utero* in rats where it remains detectable for 24 h after i.p. injection.

### MAb RVFV-268 prevents infection of dams and fetuses when administered prior to challenge

MAb RVFV-268 was remarkably effective at preventing lethal disease in adult mice when given prior to or up to 48 h after infection as previously described[23]. To determine whether mAb RVFV-268 could also protect pregnant dams and developing fetuses from infection, we used an established model of congenital RVF in SD rats[25,26]. Pregnant SD rats infected with pathogenic RVFV (strain ZH501) at E14 are susceptible to severe hepatic disease and death between 2 to 6 dpi. Vertical transmission occurs in 100% of all infected dams, even those without clinical disease, leading to increased levels of fetal mortality. High levels of infectious virus and vRNA are detectable in the placentas from infected dams demonstrating the susceptibility of this rat model to vertical transmission by RVFV.

To evaluate whether mAb RVFV-268 can provide prophylactic protection when administered prior to RVFV infection, 10 mg/kg of mAb RVFV-268, control mAb DENV-2D22, or saline was delivered i.p. to pregnant SD rats at E14 two hours prior to s.c. inoculation with $1 \times 10^5$ pfu of RVFV (Fig. 3). Euthanasia was performed at either 3 dpi (E17) or 6 dpi (E20) Liver and serum samples were collected from the dam, while the amniotic fluid, placenta and viscera were collected from all fetuses within each litter. Of the dams that were euthanized at 3 dpi, 3 out of 6 (50%) of the dams receiving mAb DENV-2D22 had already succumbed

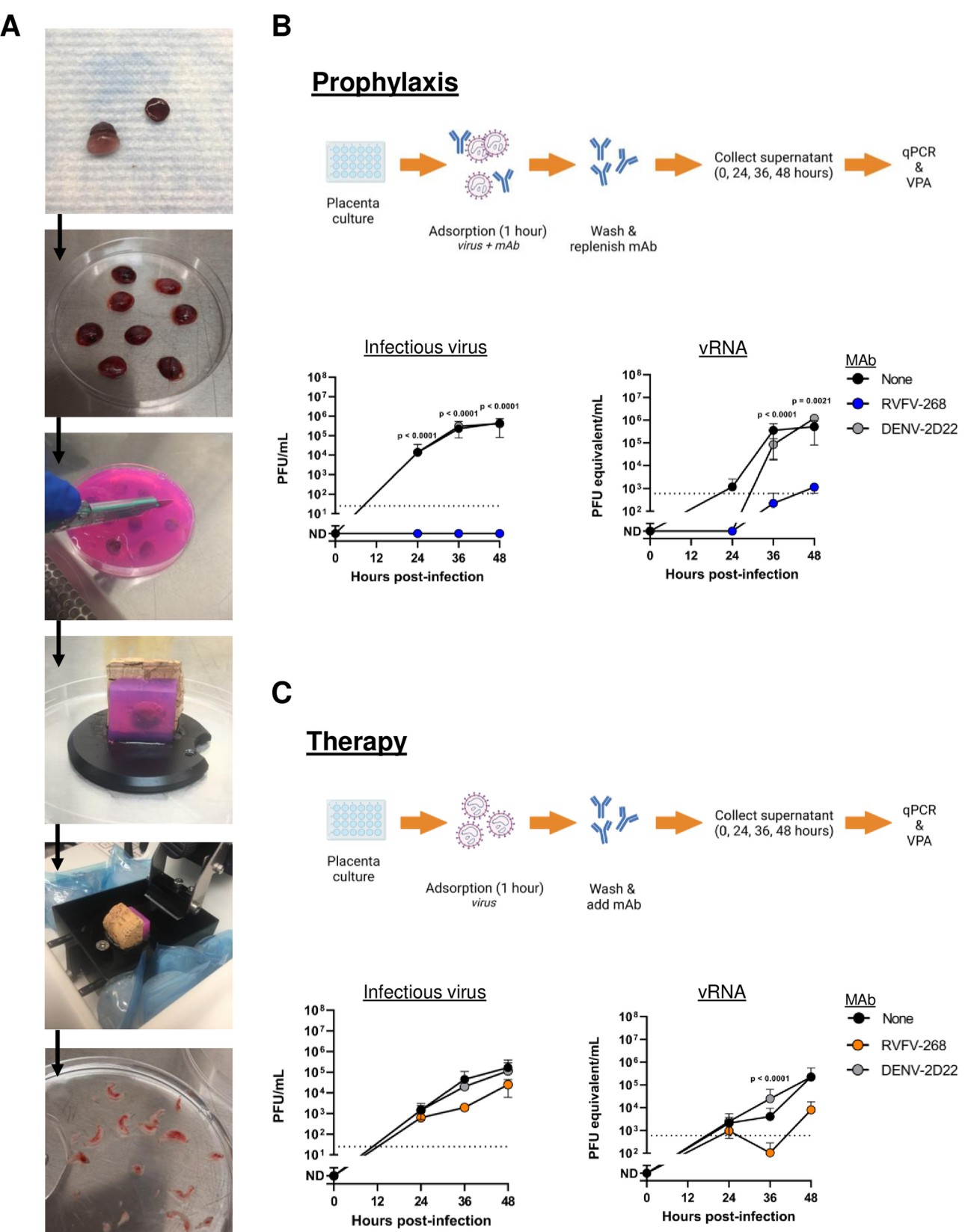

to disease, whereas 100% (*n* = 6) of those that received the mAb RVFV-268 survived to 3 dpi (Supp Figure 2A). At 6 dpi, 66.6% (4 out of 6) and 100% of the animals who received mAb DENV-2D22 or mAb RVFV-268, respectively, were still alive (Supp Figure 2B). These results demonstrate that mAb RVFV-268 administration prior to infection results in increased survival of pregnant dams.

For dams that were euthanized at 3 dpi (E17) and received the control mAb DENV-2D22, virus was found widespread throughout the dam and fetuses, as shown in our previous studies[25,26]. Infectious virus was detected in the livers from 5 out of 6 dams ($3.8 \times 10^4$ - $3.0 \times 10^8$ pfu/mL) and the serum of 4 out of 5 dams ($6.8 \times 10^3$ - $4.3 \times 10^7$ pfu/mL) (Fig. 3B, left). Serum was not collected from one dam due to severity of

**Fig. 1 | MAb RVFV-268 reduces viral replication in rat placenta explant cultures.**
**A** Placentas collected from SD rats at E14 were sliced into 300 μm sections using a vibratome. **B** For prophylaxis studies (top panel), explant slices were incubated with a combination of RVFV (1 ×10⁵ pfu) and mAb RVFV-268 or control mAb DENV-2D22 (10 ng) for 1 h, then washed and replenished with antibody (10 ng) in culture medium (*n* = 3 pairs of placenta slices from one dam). **C** For therapeutic studies (bottom panel), explant slices were incubated with RVFV (1 ×10⁵ pfu) for 1 h, then washed and treated with RVFV-268 or control DENV-2D22 antibody (10 ng) in culture medium (*n* = 3 pairs of placenta slices from one dam). Culture supernatant was collected at 0, 24, 36, 48- time-points then analyzed by viral plaque assay (bottom, left) or qPCR (bottom, right). Cultures without antibodies (none) served as a positive control. Limit of detection (LOD) = dashed line. Data are presented as mean values +/- standard deviation. Two-sided analysis of variance (ANOVA) was performed. Created with biorender.com. Raw data are presented in the Source Data file.

disease. Viral RNA was detected within the liver of all control mAb treated dams. Within the fetal samples, infectious virus was detected in the amniotic fluid from 4 out of 6 dams, the placentas from 5 out 6 litters (88.7% of all placentas), and the viscera from 5 out of 6 litters (81.8% of all fetuses). vRNA within the maternal (liver and serum) and fetal samples (amniotic fluid, placenta, and viscera) correlated with infectious virus titers, albeit some samples that did not have detectible infectious virus contained vRNA (Fig. 3B, right).

For dams receiving mAb RVFV-268, infectious virus was not recovered from dams or fetuses at 3 dpi (Fig. 3B, left). Viral RNA was detected just above the limit of detection (LOD) in amniotic fluid, placenta, and viscera from 2 out of 6 litters; vRNA was not detected in any of the samples collected from the dams. Since qRT-PCR is more sensitive than viral plaque assays, viral isolation was attempted to amplify infectious virus from raw samples from the fetuses that had low level of vRNA, but infectious virus could not be propagated as determined by qRT-PCR and by immunofluorescent staining.

When dams were euthanized at 6 dpi (E20), the number of mAb DENV-2D22 control dams in the experiment was reduced to two because the others did not survive that long. As expected in these control animals, infectious virus was found widespread throughout maternal and fetal tissues (100% of placentas; 83.3% of fetal viscera) (Fig. 3C, left), albeit approximately 1000-fold less than what was detected at 3 dpi. Infectious virus was not detected in the serum of the one dam from which we could collect a serum sample. Similar levels of vRNA were detected in these samples as well (Fig. 3C, right). In contrast, dams receiving mAb RVFV-268 did not have infectious virus (Fig. 3C, left) or detectable vRNA (Fig. 3C, right) within any of the samples collected from the dams or fetuses. The lack of infectious virus and vRNA at 6 dpi (E20) suggests that the low levels of vRNA detected at 3 dpi might be from residual RNA from viral debris (Fig. 3B, right). Thus, compared to the DENV control mAb, mAb RVFV-268 effectively prevented infection of the dams and fetuses when given prior to inoculation with RVFV.

Severe liver disease is a common outcome of RVFV infection in rodents. The lack of RVFV vRNA and infectious virus in the liver (Fig. 3) suggests sterilizing immunity was achieved in dams prophylactically treated with mAb RVFV-268. Corroborating this finding, more severe histopathology was observed in the livers of mAb DENV-2D22 treated animals at 3 dpi, compared to those that received mAb RVFV-268 (Fig. 4, middle and right panels). Livers from dams who received control mAb demonstrated histopathologic changes commonly associated with RVFV infection, including foci of hemorrhage, necrosis, and inflammation. In contrast, livers from dams who received mAb RVFV-268 demonstrated only mild inflammation. Immunofluorescent staining of livers collected at 3 dpi showed RVFV nucleoprotein throughout the liver of control mAb treated dams, whereas antigen was not seen in the livers of dams who received mAb RVFV-268 (Fig. 4, left panels).

### Protection associated with prophylactic delivery of mAb RVFV-268 is dose-dependent
Because a dose of 10 mg/kg of mAb RVFV-268 provided apparent sterilizing protection for the dams and fetuses, we performed a dosing study to determine the lowest effective dose that can protect from

infection. Prophylactic delivery of 10 mg/kg or 0.5 mg/kg of mAb RVFV-268 protected 100% or 90% of mice from lethal hepatic disease, respectively, in a previous study[23]. Dams at E14 were i.p. injected with 1 mg/kg or 0.1 mg/kg of mAb RVFV-268 two hours prior to infection; these doses are 10- and 100-fold less than what was delivered in Fig. 3, respectively. Three days after s.c. injection with 1 ×10⁵ pfu of RVFV, the dams were euthanized, and samples were collected from the dams and fetuses as performed previously (Fig. 5A). All dams who received 1 mg/kg of mAb RVFV-268 survived to 3 dpi, whereas 1 out of 6 dams who received 0.1 mg/kg of the antibody was euthanized at 2 dpi (Supp Figure 2C).

Of the six dams who received 1 mg/kg of mAb RVFV-268, only one became infected with RVFV as determined by the presence of infectious virus in its liver at 3 dpi. Although virus was detected in the liver, none of the other tissues from this dam, including fetal-associated tissues, had detectable infectious virus. The remaining five dams and their matched fetal tissues also did not contain infectious virus (Fig. 5B, left). Viral RNA, however, was detected in the amniotic fluid from the infected dam. Furthermore, one dam without detectable infectious virus in dam or fetal tissues had viral RNA in the placenta (2/17) and viscera (1/8) which may be residual RNA from degraded input virions, as seen in the previous experiment (Fig. 5B, right). When dams received 0.1 mg/kg of mAb RVFV-268, infectious virus was detected in the liver of 2 out of 6 dams, the serum of 1 out of 5 dams, and the amniotic fluid of 1 out of 6 dams at 3 dpi (Fig. 5B, left); similar concentrations of viral RNA were also detected (Fig. 5B, right). These results suggest that prophylactic delivery of at least 1 mg/kg of mAb RVFV-268 protected the developing fetuses from exposure to infectious virus by 3 dpi. The dam, however, may still experience infection peripheral to the conceptus.

### MAb RVFV-268 can protect the dam and fetus when delivered as late as 24 h post-challenge
To determine whether delivery of mAb RVFV-268 after RVFV challenge can protect the dam and fetus from infection, dams were injected with 10 mg/kg of antibody 6 or 24 h after inoculation with 2 x 10⁵ pfu of RVFV. Three days after challenge, the dams were euthanized and samples were collected (Fig. 6A).

For dams that received antibody 6 h after infection, neither infectious virus nor vRNA was detected in any of the samples from the dams that received mAb RVFV-268. Similarly, fetus-associated samples did not have infectious virus or vRNA (Fig. 6B). Both dams that received mAb DENV-2D22 had detectible vRNA in the liver, serum, and amniotic fluid. All but two placentas had detectible vRNA.

When mAb RVFV-268 was delivered 24 h after RVFV challenge, infectious virus and vRNA was also not detected in the dam samples. However, low levels of vRNA, but not infectious virus, were detected in several placenta from 2 out of 5 of the mAb RVFV-268 treated dams (Fig. 6C, right). In contrast, both dams that received mAb DENV-2D22 had detectible vRNA in all samples except for the viscera of one pup, and most samples also had detectable infectious virus (Fig. 6C). Thus, delivery of mAb RVFV-268 as late as 24 h after inoculation with RVFV can remarkably reduce both maternal and fetal infection.

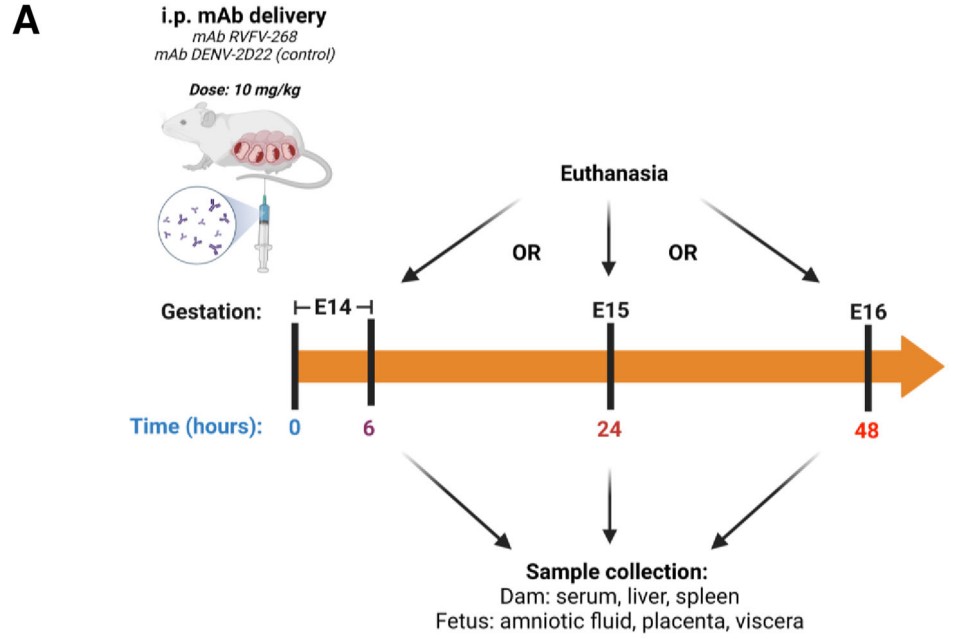

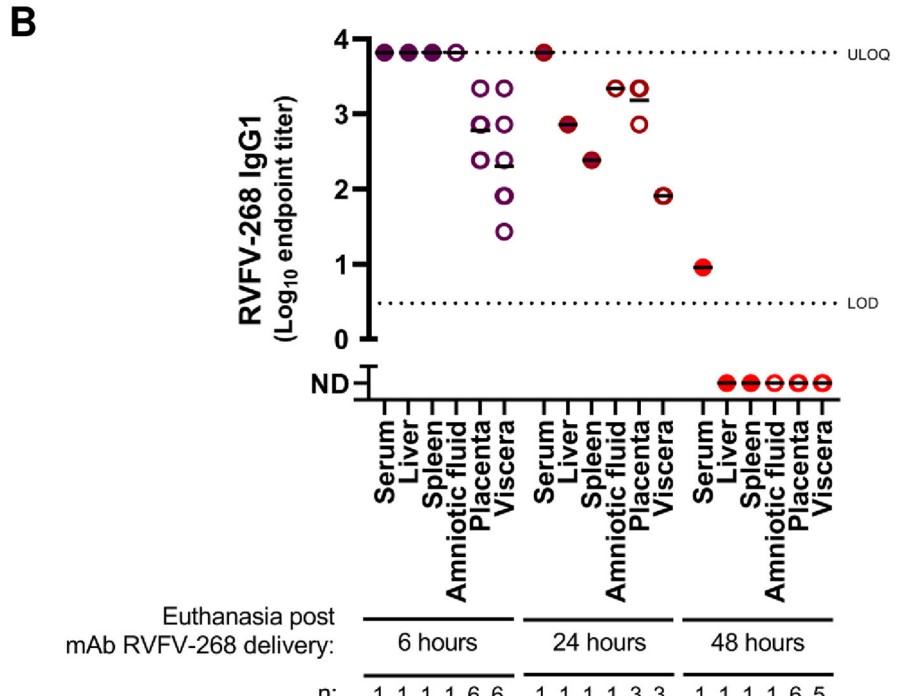

**Fig. 2 | MAb RVFV-268 crosses the placental barrier and enters the fetus. A** SD rats (E14) were intraperitoneally injected with 10 mg/kg of mAbs RVFV-268 or DENV-2D22 (*n* = 3). 6, 24, or 48 h later one dam was euthanized, and samples were collected from the dam (serum, liver, spleen; solid circles) and fetus (amniotic fluid, placenta, viscera; open circles). **B** RVFV Gn-specific human IgG1 ELISA semi-quantified based on $\log_{10}$ endpoint titer. Black lines = mean. LOD or upper limit of quantification (ULOQ) = lower and upper dashed lines, respectively. Created with biorender.com. Raw data are presented in the Source Data file.

## Discussion

RVFV continues to cause devastating diseases in livestock and humans in Africa and the Middle East. Despite the use of live-attenuated vaccines (LAV) in livestock within endemic regions to control animal disease, multiple human and animal populations remain prone to RVFV infection. The spread of RVFV to non-endemic or naïve populations is feasible due to continued agricultural trade[34,35] and the risk of mosquito population expansion due to climate change[36]. Even within endemic regions, LAVs are not recommended for pregnant animals due to the risk of LAV vertical transmission that may cause developmental complications for the fetus[15]. As for humans, there are no FDA-approved vaccines, although some vaccine candidates are currently in pre-clinical or early-stage clinical trials[37]. Despite significant strides being made in vaccine development, safety and efficacy studies in pregnant women are often conducted near the end of the clinical trial process or omitted altogether. These limitations emphasize the need for more preclinical research into RVFV therapeutics particularly for pregnant populations.

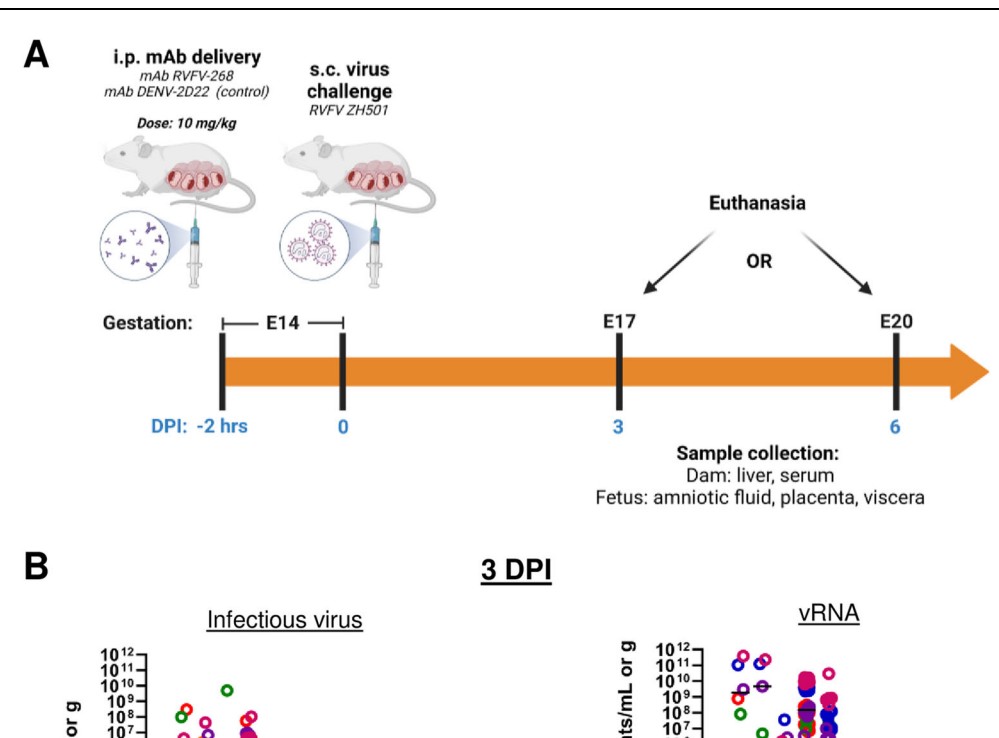

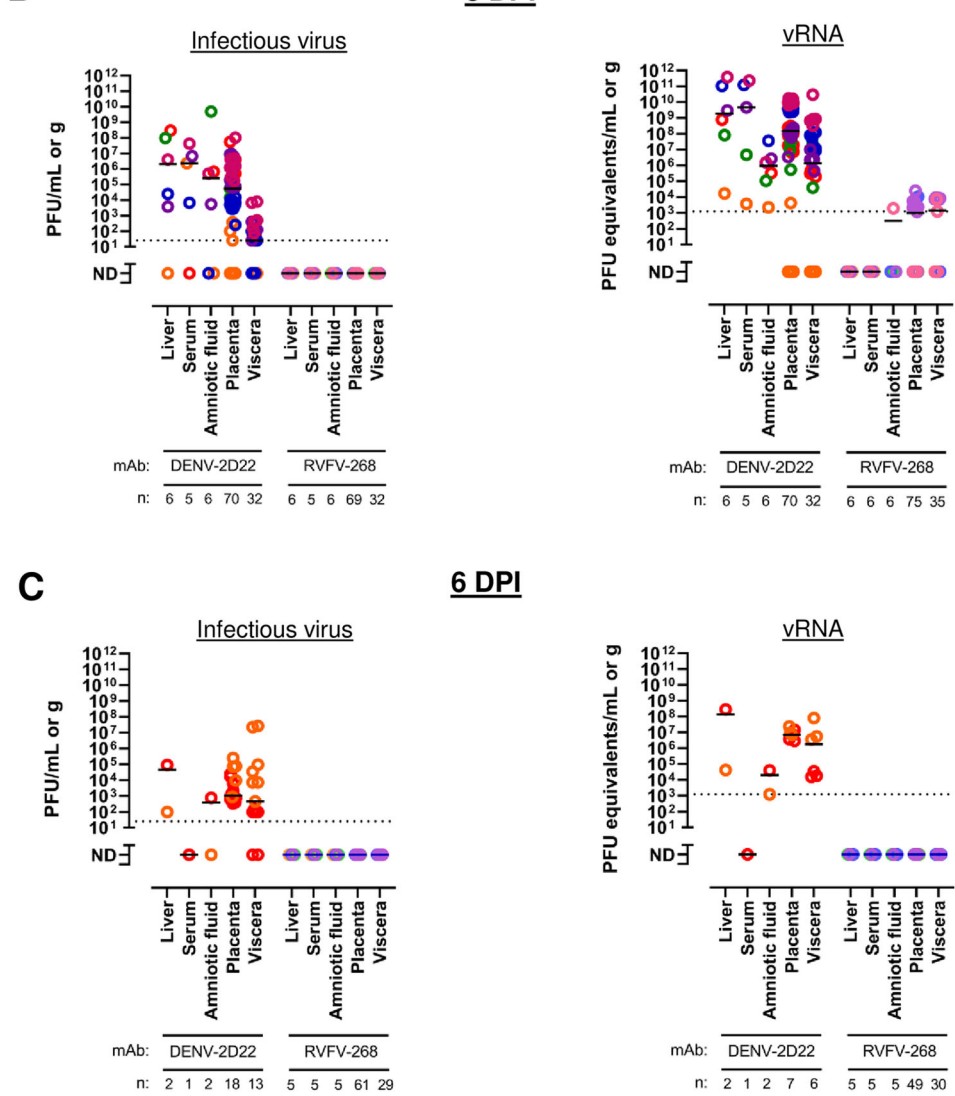

**Fig. 3 | Prophylactic delivery of mAb RVFV-268 protects the dam and fetus from infection. A** SD rats (E14) were injected i.p. with 10 mg/kg of mAbs RVFV-268 or control DENV-2D22 2 h prior to s.c challenge with RVFV (2 ×10⁵ pfu). Dams were euthanized at **B** 3 dpi or **C** 6 dpi and samples were collected from the dam (liver, serum) and fetus (amniotic fluid, placenta, viscera), then analyzed by viral plaque assay (left) or qPCR (right). Each color data point represents samples collected from a single dam and its litter. Black lines = mean. Limit of detection (LOD) = dashed line. Created with biorender.com. Raw data are presented in the Source Data file.

MAbs can effectively treat infectious diseases in vivo[38] and some promising candidates are either in clinical trials or were given Emergency Use Authorization for treatment of COVID-19 by the Food and Drug Administration[39]. Furthermore, mAb therapeutics are regularly used during pregnancy[40]. The development of mAbs for therapeutic intervention has gained popularity in the bunyavirus field. Multiple studies have evaluated either Fc-mediated immune functions[41] or neutralizing capabilities[23,24,42–44] of mAbs. Although many neutralizing mAb significantly reduce disease in vivo, their protective efficacies had only been evaluated in the context of hepatic or neurological disease in non-pregnant mice. Here, we evaluated the potent neutralizing human mAb RVFV-268 for its ability to protect pregnant rats from RVF and to limit vertical transmission of RVFV to fetal tissues.

Chapman, et al.[23] demonstrated that mAb RVFV-268 reduced the rate of lethal hepatic disease in adult mice when delivered prophylactically or as a treatment 2 or 4 days after infection[23]. In that study, sterilizing immunity against RVFV infection was achieved when the mAb was delivered 2 h prior to inoculation. In our studies, multiple experimental parameters indicated that sterilizing immunity also occurred when mAb RVFV-268 was administered 2 h before inoculation of pregnant rats with RVFV. These data include the lack of infectious virus, vRNA, and immunofluorescent staining for RVFV antigen within maternal and fetal samples. In some cases, only low levels of vRNA were detected which could be due to the presence of neutralized virus or degraded virus RNA remaining in the tissues or still circulating in the blood. Future studies using more sensitive assays would aid in understanding the full extent of protection provided by mAb RVFV-

268 in pregnant rats. This approach would include studies allowing the infected and treated dams to deliver their fetuses, measuring if any virus is in the fetuses at delivery, and monitoring fetuses for health and development. Seroconversion of the dams several weeks after infection may indicate whether true sterilizing immunity was achieved. Regardless, even when sterilizing immunity was not achieved in the presented studies, as in the dose-down experiment, viral titers in the liver and serum were significantly reduced in mAb RVFV-268 treated dams compared to control mAb-treated dams, leading to even lower viral titers in fetal tissues. These lower titers also were associated with reduced immunofluorescent staining and pathology in the liver. Thus, even in cases of breakthrough infection with insufficient protection provided by mAb RVFV-268, low levels of mAb RVFV-268 reduced viral burden within the dams and fetuses and reduced liver pathology within the dams.

Remarkably, dams and fetuses were protected when mAb RVFV-268 was administered as late as 24 hpi, and this speaks to the potent activity of this neutralizing antibody. Dams and fetuses had no detectable vRNA or infectious virus when the mAb was given 6 h after infection, and only a small amount of breakthrough vRNA was detected in placenta from dams given the antibody 24 h after infection. Additional studies can further explore the latest time frame that mAb RVFV-268 administration could provide complete or partial protection of the dams and fetuses from vertical transmission of RVFV.

MAb RVFV-268 used in these studies is a recombinant human IgG1 protein. IgG1 is known to bind with high affinity to the neonatal Fc receptor, FcRn, which is found throughout the body and mediates passive antibody delivery between mother and child in utero[45,46]. Binding of antibodies to FcRn and subsequent endocytosis dictates the concentration of circulating antibodies. FcRn molecules are highly expressed on vascular endothelium which is considered the main site of IgG cycling. The binding affinity between the Fc of one species and FcRn of another species varies[47] (herein, species incompatibility), and as a result can impact antibody cycling and serum concentrations. Despite species incompatibility between the human Fc and rat FcRn, mAb RVFV-268 was transferred across the placenta and into the fetuses, which is consistent with other reports of Fc-FcRn species incompatibilities[48]. Interestingly, however, the mAb RVFV-268 quickly cleared from the maternal and fetal tissues as early as 48 h after delivery. The kinetics of rapid clearance of mAb RVFV-268 may be related to the increased expression levels of FcRn in these pregnant animals. Further studies directly comparing the clearance rate of mAb RVFV-268 in pregnant and nonpregnant animals are warranted. General features of this antibody, such as paratope charges and isoelectric point may also contribute to the clearance rate observed in this model[49,50]. Rational selection and design of neutralizing antibodies is important to consider when developing mAb therapies. Additional engineering options may improve application of mAb RVFV-268 during pregnancy. For instance, increasing vertical transfer of antibodies from the mother to fetus in utero would provide more protection to the fetus than only the indirect protection conferred by reducing infection and replication in the mother. Introducing three amino acid mutations, M252Y/S254T/T256E (i.e., YTE mutation), within the Fc region of human IgG antibodies increases FcRn binding[51], suggesting a mutation of this kind may also improve vertical transfer of the antibody in utero, which could be tested empirically. Although improving transplacental transfer of antibodies by inserting the YTE mutation and altering antibody-FcRn binding affinity may be species-dependent, it does improve the transfer of human mAbs across the placenta of nonhuman primates[52]. In addition to examining the impact of other mutations on antibody transfer[53,54], future studies in which these animals have the human FcRn introduced or where mAb RVFV-268 is grafted onto a rat Fc backbone may be warranted.

In addition to its neutralizing capabilities, antibody-dependent enhancement (ADE) is a concern when employing passive antibody

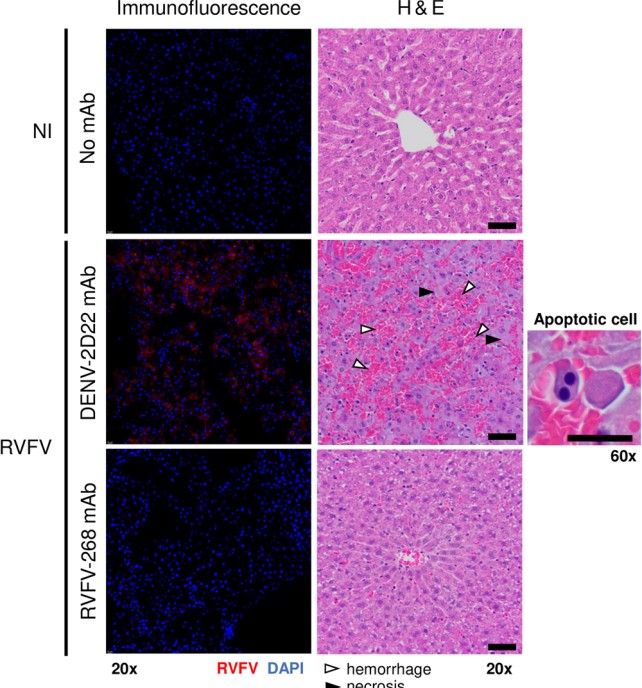

**Fig. 4 | Dams prophylactically treated with mAb RVFV-268 had minimal liver pathology and infection.** Immunofluorescent (left) and H & E (middle) images from dams prophylactically treated with mAbs RVFV-268 (n = 2) or DENV-2D22 (n = 2) prior to inoculation with RVFV. Dams were euthanized and livers were harvested at 3 dpi. For the immunofluorescent images, RVFV nucleoprotein staining is in red and nuclei staining is in blue. For the H & E images, white and black arrows highlight areas with hemorrhage and necrosis, respectively. 20x images, scale bar = 50 μm. Panels are representative pictures of a liver from each treatment group from Fig. 3. Livers from uninfected dams (n = 2) not treated with mAb were controls. A representative image of an apoptotic cell (right) from an H & E stain. 60x image, scale bar = 20 μm.

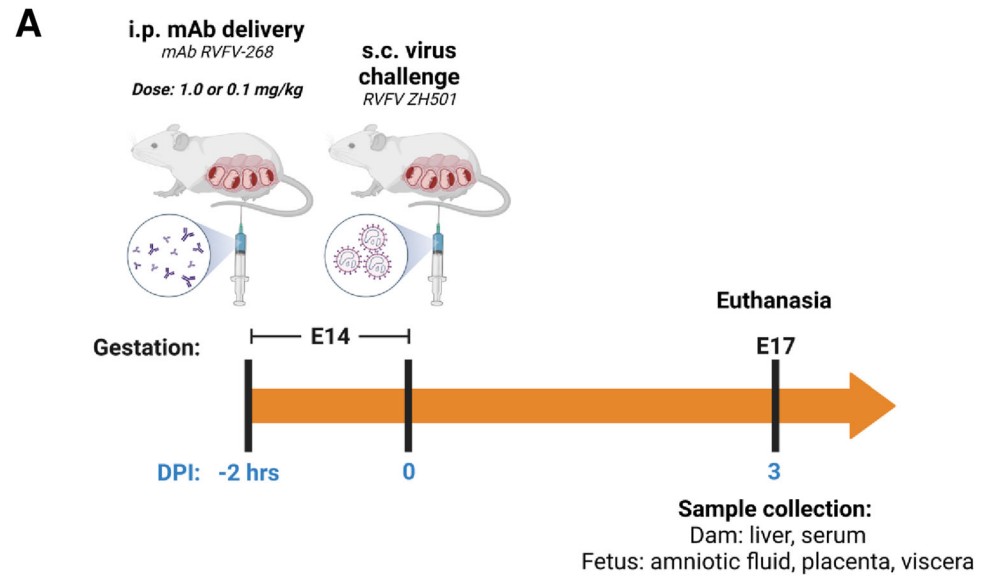

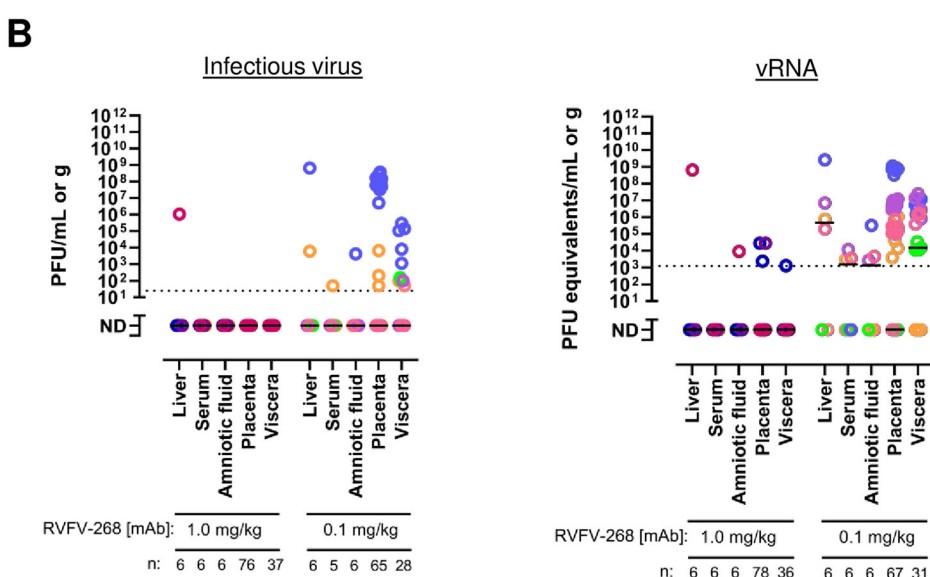

**Fig. 5 | Prophylactic protection with mAb RVFV-268 is dose-dependent. A** SD rats (E14) were injected i.p. with 1 mg/kg or 0.1 mg/kg of mAb RVFV-268 2 h prior to challenge with RVFV (2 ×10⁵ pfu). Dams were euthanized at 3 dpi, samples were collected from the dam (liver, serum) and fetus (amniotic fluid, placenta, viscera), then analyzed by **B** Viral plaque assay (left) or qPCR (right). Each color data point represents samples collected from a single dam and its litter. Black lines = mean. Limit of detection (LOD) = dashed line. Created with biorender.com. Raw data are presented in the Source Data file.

therapies especially since tissue resident cellular immunity may not be present in naïve individuals. ADE is a complex set of phenomena where previously produced or passively acquired antibodies that recognize surface-expressed virus antigens can enhance disease. These disease enhancements may be by facilitating entry into a host cell by means other than virus-host receptor binding[55], such as an Fc receptor. Our results do not indicate enhanced vertical transmission with mAb RVFV-268 intervention.

Further insights into human pathogenesis of RVFV in the context of pregnancy are needed to refine the animal models. Tissue specific immune cell presence or co-infection with other pathogens is a critical consideration in the context of viral infection, and moreover, the complexity of individual inflammatory profiles creates a difficult situation to resolve. For example, recent data pertaining to SARS-CoV-2 infection in humans suggests that FcγRIIIa may provide an entry route into alveolar macrophages and subsequent inflammasome activation especially for antibodies with an afucosylated glycan on the Fc domain[56–58]. Since placental macrophages are present in pregnancy and

express Fc receptors[59], pathways, such as the aforementioned one, should be explored in the context of RVFV infection and monoclonal antibodies as a potential avenue of increased undesired inflammation. If needed, however, ADE could be prevented by introducing mutations that decrease or ablate antibody effector functions by introducing the LALA-PG or TM mutations[60,61]. Ultimately, the Fc profile of these antibodies and immune cells will need careful consideration given the sensitive nature of pregnancy and the tissue involved during pregnancy.

Mab therapy is being pursued as an avenue to prevent vertical transmission of other infectious diseases, including herpes simplex virus (HSV), which could provide insight into emerging viruses like RVFV. The importance maternal antibodies play in protecting neonates from vertical transmission of HSV was first realized after clinical evidence found babies from women with primary HSV infection were more likely to become infected or succumb to severe disease, compared to babies born to HSV-infected women who had recurrent episodes[62,63]. Indeed potent neutralizing HSV antibodies were later

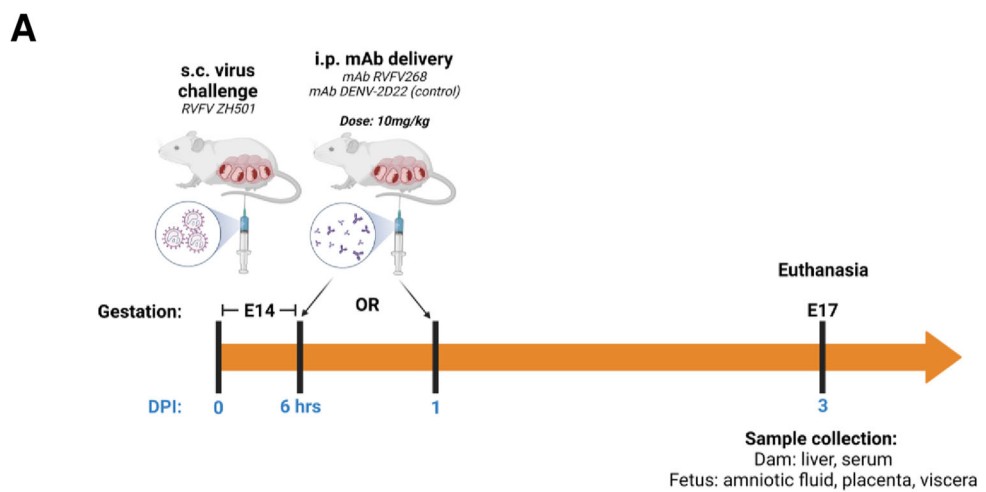

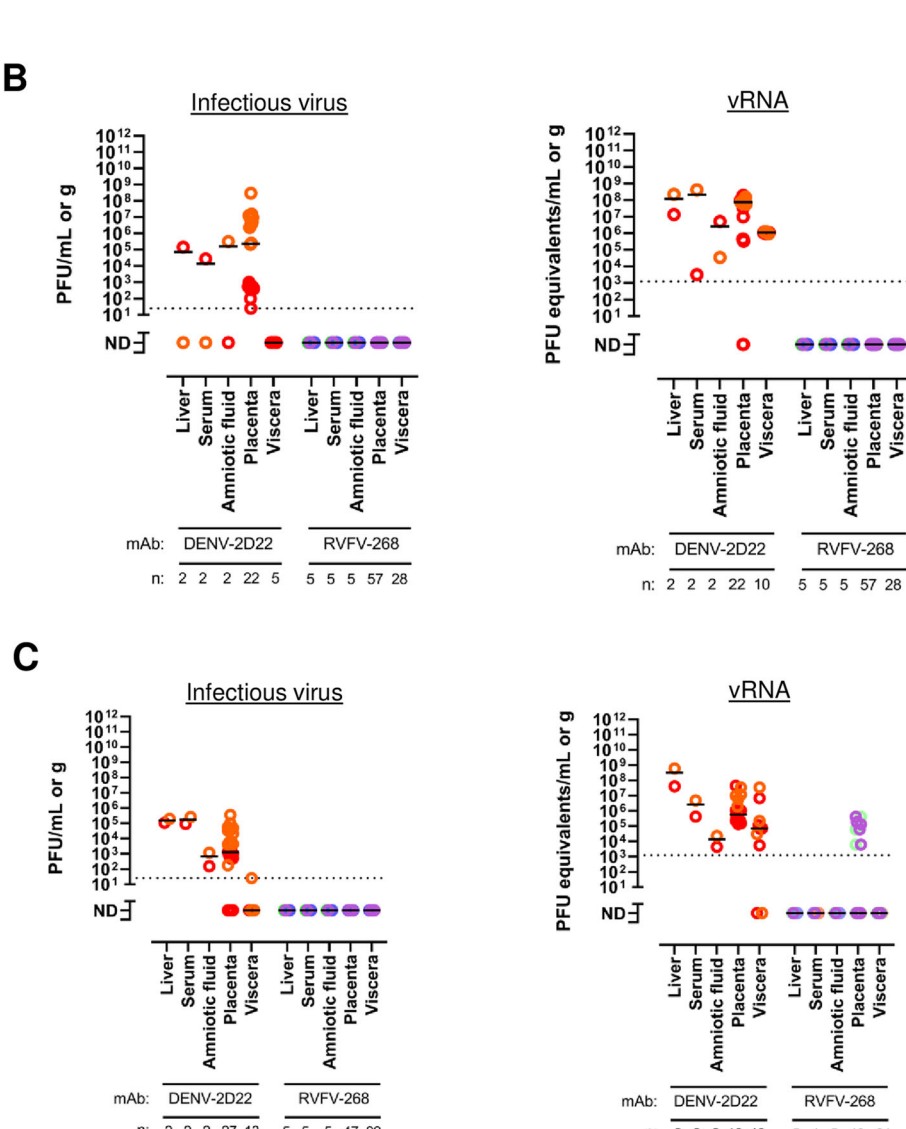

**Fig. 6 | Therapeutic delivery of mAb RVFV-268 can protect dams and fetuses from infection when administered as late as 24 h post-challenge. A** SD rats (E14) were injected i.p. with 10 mg/kg of mAbs RVFV-268 or DENV-2D22 6 or 24 h post-challenge with RVFV (2 ×10⁵ pfu) then euthanized at 3 dpi. Samples were collected from the dam (liver, serum) and fetus (amniotic fluid, placenta, viscera). **B** Samples from dams treated with mAb 6 h post-RVFV challenge were analyzed by viral plaque assay (VPA) (left) or qPCR (right). **C** Samples from dams treated with mAb 24 h post-RVFV challenge were analyzed by VPA (left) or qPCR (right). Each color data point represents samples collected from a single dam and its litter. Black lines = mean. LOD = dashed line. Created with biorender.com. Raw data are presented in the Source Data file.

detected in the cord blood and serum of babies born to HSV immune women, indicating passive transfer of maternal antibodies likely provides protection[64]. In the laboratory, passive administration of HSV immune sera or maternal vaccination protected neonate mice from vertical transmission of HSV[64]. Passive delivery of human mAbs intravenously[65] or applied topically to the vagina[66] of pregnant mice infected with HSV reduced neonatal infection or death further demonstrating the therapeutic potential of mAbs against vertically transmitted diseases. A topically applied vaginal film containing HSV-1 and HSV-2 mAbs recently completed phase I clinical trials[67]. This combination antibody therapy, consisting of antibodies that conferred protection in mice[66], could play an important role in preventing perinatal HSV infection, the most common cause of vertically transmitted HSV. Similar experimental approaches may provide additional understanding of vertical transmission mechanisms of RVFV.

These data provide evidence that mAb RVFV-268 should be further investigated as a potential prophylactic and/or therapeutic against RVF. As evidenced by the effective combination of pre-exposure and post-exposure antibody treatments for addressing the COVID pandemic, neutralizing Ab therapeutics are considered an important armament in the battle against emerging viral diseases, and the ability to use them after infection to alleviate symptoms or disease severity is essential. MAbs or combinations of potent antibodies that could be used in humans during a potential outbreak or upon the introduction of RVFV to naïve regions would be beneficial to mitigate the potential negative outcomes in a population. Future clinical studies may consider testing the use of pre-exposure mAb in vulnerable populations, such as those who are or plan to become pregnant, as a potential countermeasure. Furthermore, the use of tools to predict RVFV outbreaks in Africa, such as the ability to predict regional outbreaks based on weather patterns may inform how, when, and where to distribute antibody pre-exposure prophylactics to vulnerable populations[68]. This study is the first to test mAb RVFV-268 in pregnant animals, and our results suggest that it is effective at preventing both maternal and fetal infection in this rodent model when given before or just after infection.

## Methods

### Ethics and biosafety

Our research complies with all relevant ethical regulations. All animal work described here was carried out in strict accordance with the Guide for the Care and Use of Laboratory Animals of the NIH and the Animal Welfare Act. The protocol was approved and overseen by the University of Pittsburgh Institutional Animal Care and Use Committee. The Association for Assessment and Accreditation of Laboratory Animal Care has fully accredited the University of Pittsburgh.

Work with virulent RVFV was performed at biosafety level 3 (BSL-3) in the University of Pittsburgh Regional Biocontainment Laboratory (RBL) following all safety precautions[25]. The University of Pittsburgh RBL is registered with the Centers for Disease Control and Prevention and the U.S. Department of Agriculture for work with virulent RVFV.

### Virus and cell culture

Vero E6 cells (CRL-1586, ATCC) were cultured in DMEM (Dulbecco's Modified Eagle Medium, Corning) supplemented with 10% (v/v) fetal bovine serum (FBS), 1% L-glutamine and 1% penicillin/streptomycin (pen-strep). Cells were maintained in a humid incubator at 37 °C with 5% $CO_2$.

Virulent RVFV strain ZH501 was generated from reverse genetics plasmids[69] and provided to our laboratory by B. Miller (CDC, Ft. Collins, CO) and S. Nichol (CDC, Atlanta, GA). RVFV was propagated on Vero E6 cells (ATCC, CRL-1587) following standard methods, and the stock titer was determined by standard viral plaque assay (VPA). Prior to animal and explant studies, stock virus was diluted in D2 medium (DMEM, 2% (v/v) FBS, 1% L-glutamine, and 1% penicillin-streptomycin)

to the desired inoculum dose. All uninfected control cohorts were mock infected with D2 medium to account for components within the medium.

### Antibodies and proteins

Recombinant mAbs (RVFV-268 and DENV-2D22) were expressed in Expi293F cells and purified by affinity chromatography[23] Recombinant RVFV Gn used for ELISAs (see below) was produced in SF9 cells using a gene encoding the ectodomain of RVFV Gn (GenBank accession number JQ068142.1 [https://www.ncbi.nlm.nih.gov/nuccore/JQ068142.1], residues 154-469)[23,24,70]. The Gn recombinant protein was purified by metal affinity chromatography on HisTrap Excel columns (Cytiva).

### Animals

Time-mated SD female rats (6–8 weeks old) were obtained from Envigo Laboratories (Inotiv). Sighting of a copulation plug verified pregnancy status and timing. Pregnant rats were delivered to the RBL and housed up to three to a cage in temperature-controlled rooms with 12 h day/12 h night light schedule. Food (IsoPro Rodent 3000) and water were provided ad libitum. For all procedures, rats were anesthetized by inhalation of vaporized isoflurane (IsoThesia, Henry Schein). Rats were implanted with programmable temperature transponders (IPTT-300, BioMedic Data Systems) s.c. between the shoulder blades.

For studies evaluating the passive transfer of antibody from dam to fetus, pregnant SD rats (E14) were anesthetized prior to intraperitoneal (i.p.) delivery of 10 mg/kg of mAb RVFV-268 or DENV-2D22 (n = 3 each). One dam from each antibody treated group was euthanized at 6, 24, or 48 h after antibody delivery. One PBS treated dam was euthanized 48 h post-treatment to serve as a negative control.

For all studies, antibodies were delivered by i.p. injection of 500 μL of the corresponding antibody (RVFV-268 (n = 5–6) or DENV-2D22 (n = 2–6)) and concentration (10, 1.0, or 0.1 mg/kg). MAb was delivered 2 h before infection for prophylactic studies or at 6 and 24 h after infection for therapeutic studies.

For RVFV challenge studies, rats were inoculated subcutaneously (s.c.) in the hind flank on embryonic day 14 (E14) with 200 μL of RVFV (2 x $10^5$ pfu) diluted in D2 medium. Body temperature and weight were recorded daily. Each animal was closely monitored twice daily for clinical signs of disease. Animals that met euthanasia criteria were humanely euthanized following standard IACUC procedures[25]. Planned euthanasia for surviving animals occurred at 3 or 6 dpi.

Upon necropsy, the following samples were collected from the dams: liver, serum, and spleen, and from the fetuses: amniotic fluid, placenta, and viscera. Liquid samples were immediately stored at −80 °C before use for downstream analyses. For tissue samples, half of the tissues were immediately frozen at −80 °C for virological analyses, while the other half was fixed in fresh 4% paraformaldehyde (PFA; Sigma) for histological analyses.

Sex differences were not analyzed in this study given that it focuses on the prevention of vertical transmission of RVFV with monoclonal antibodies. Only pregnant female rats were pertinent to the study.

### Explant studies

Pregnant SD rats at E14 were euthanized, and placentas were collected at necropsy and immediately stored in room temperature D2 medium. Within 30 min, placentas were embedded in HBSS with 4% agarose. Placentas were sliced using a Leica VT100S vibratome with a ¼ inch cube of agarose supporting the outside of the placenta. The embedded placentas were mounted to the vibratome, and the collection basin was filled with chilled PBS. 300 μm transverse sections were cut with a double edge persona blade (Electron Microscopy Sciences) with the frequency set at 9 and amplitude set at 1. One placenta section of comparable size and composition were placed in each well of a 24 well plate containing 500 μL of D2 medium until infection.

For prophylactic studies, three placentas were collected from one dam. Two slices from each placenta (n = 6 slices per group) were distributed to each therapeutic group explained below. D2 medium was removed from each placenta section prior to inoculation with a combination of RVFV ($1 \times 10^5$ pfu) and corresponding antibody (10 µg, 1 µg, 100 ng, or 10 ng). 200 µL of inoculum and mAb was incubated for a 1 h adsorption period, the inoculum containing antibody was removed, washed one time with PBS, and then 1 mL of D2 medium containing the corresponding antibody (10 µg, 1 µg, 100 ng, or 10 ng) was added to the corresponding placenta samples. Samples without the antibody served as a positive control, while samples without virus served as a negative control. The plate was cultured at 37 °C in 5% $CO_2$ for 48 h. Supernatant from each placenta was collected and pooled at 0, 24, 36 and 48 hpi, resulting in three biological replicates per group. Pooled supernatant was stored at −80 °C until further analyses by qRT-PCR and VPA.

For treatment studies, three placentas were collected from one dam. Two slices from each placenta (n = 6 slices per group) were distributed to each treatment group explained below. D2 medium was removed from each placenta section prior to inoculation with RVFV ($1 \times 10^5$ pfu). Following a 1 h adsorption period, the inoculum was removed, washed once with PBS, and then 1 mL of D2 medium containing 1 ng of antibody was added to the corresponding placenta samples. The plate was cultured for 48 h then the same time-points and pooled supernatant samples were collected and analyzed as above.

## Viral plaque assay

Viral titers were determined by standard viral plaque assay. 200 µL of tissue homogenates, serum, or amniotic fluid were serially titrated onto monolayers of Vero E6 cells (CRL-1586, ATCC). After an hour incubation for viral adsorption, the cellular monolayers were washed once with PBS, then 3 mL of agarose overlay (minimum essential medium, 2% FBS, 1% penicillin/streptomycin, 1.5% (1 M) HEPES buffer, and 0.8% SeaKem agarose) was applied and plates were incubated at 37 °C, 5% $CO_2$ for three days. Cells were fixed with 37% formaldehyde, then stained with crystal violet and quantified using standard methods. For plaque assay data, none detected (ND) is defined as anything below the LOD (determined based on lowest number of countable plaques).

## RNA isolation and qRT-PCR

Tissues were homogenized in D2 medium (w/v), then stored at −80 °C until downstream processing. For RNA isolation, 100 µL of tissue homogenates, serum, or amniotic fluid were inactivated in 900 µL of Tri-Reagent (Invitrogen) then RNA was isolated using the Invitrogen PureLink Viral RNA/DNA kit[25]. One-step qRT-PCR targeting the RVFV L segment was performed to quantify vRNA[25]. For qRT-PCR data, none detected (ND) is defined as anything below the LOD (determined based on lowest standard curve titer that amplifies).

## Histopathology and immunofluorescent microscopy

Hematoxylin and eosin (H & E) staining was performed on fixed tissues that were processed and paraffin embedded (PFE) by the University of Pittsburgh McGowen Institute Histology Core. PFE tissues were sliced to 5 µm sections using the Histocore Autocut automated rotary microtome (Leica), placed on charged slides, then stained by standard H & E methods. Microscopic histopathologic examination was performed by a board-certified anatomic pathologist, blinded to treatment groups.

For immunofluorescent imaging, fixed tissues underwent tissue preparation and cryopreservation through 24 h incubations in 20% sucrose in PBS, then 40% sucrose in PBS prior to freezing in mounting optimal cutting temperature (OCT) compound (Fisher

Healthcare) and storage at −80 °C. Frozen tissues were sliced to 5 to 7 µm sections using the CryoStar X7- CryoStat (Thermo Fisher Scientific) and stored at −80 °C until the staining procedure was performed. The cryo-sections were rehydrated with PBS containing 0.5% bovine serum albumin (PBB), and then blocked with 5% normal goat serum in PBB. The tissues were probed with custom rabbit anti-RVFV nucleoprotein polyclonal antibody (1:50 dilution; Genscript) followed by a secondary antibody, goat anti-rabbit IgG, labeled with Alexa Fluor594 (1:1000 dilution; Invitrogen A11012). Following counterstaining with 4′,6-diamidino-2-phenylindole (DAPI), a coverslip was mounted with gelvatol. Primary-delete and no-infection controls were used for imaging. For immunofluorescent staining of Vero E6 cells on coverslips, similar staining parameters were performed as above, except for blocking with 10% normal goat serum (Invitrogen).

H & E stained images were obtained using an Olympus CX41 microscope with a Levenhuk M30 base attachment. Immunofluorescent images were obtained using Nikon A1 confocal microscope provided by the University of Pittsburgh Center for Biological Imaging. Denoising and contrast were formed using Fiji with Image J. Representative images are shown for each group including images from an uninfected and untreated dam as a control.

## ELISA

For semi-quantitation of RVFV Gn-specific antibodies within dam and fetal samples, tissues were homogenized w/v in D2 medium. Maxisorp ELISA plates (Thermo Fisher Scientific) were coated with 75 ng of recombinant Gn per well overnight at 4 °C. The next day, plates were washed 3 times in 0.1% PBS-Tween (PBST), blocked with 5% milk in 0.1% (PBST for 1 h at 37 °C, and then 100 µL of 1:3 serial dilutions of each sample was plated in duplicate at 37 °C for 2 h. Next the plates were washed 3 times with PBST, and then 100 µL of goat anti-human-HRP conjugated IgG [200 ng/mL; 1:5000 dilution] (Invitrogen; 62-8420) was added to each well and incubated at 37 °C for 1 h. Plates were washed 3 times with PBST, and then 100 µL of 3, 3′, 5, 5′ – tetramethylbenzidine substrate (Sera Care; 5120-0047) was added per well. After the plates were incubated at room temperature for 15 min, 50 µL of stop solution (Sera Care; 5120-0021) was added per well before the optical density (OD) was read at 450 nm. Background signal was subtracted from the signal for each antibody-treated group and was calculated as the OD + (2*standard deviation) of each PBS control sample (serum, liver, spleen, amniotic fluid, placenta, and fetus viscera). When multiple samples were obtained for a tissue (i.e., placentas or fetal tissues) the highest value within the PBS control group was selected as the background. The lowest and highest dilutions measured (1:3 and $1:3^8$) that had positive OD values above background were considered the limits of detection (LOD) and upper limit of quantitation (ULOQ) for this assay, respectively.

## Virus growth assay

Vero E6 cells (ATCC, CRL-1587) were seeded on L-lysine (R&D Systems) treated coverslips placed within 24 well tissue culture plates. Twenty-four hours after seeding, the cells were inoculated with 50 µL of liquid samples or tissue homogenates in 500 µL of D2 medium. Cells were inoculated with samples from infected or uninfected controls. Cells were monitored daily for signs of cytopathic effects compared to uninfected controls. After 96-hours of incubation at 37 °C in 5% $CO_2$, supernatant was collected, and the cells were fixed with 4% PFA. Supernatant also was collected at 0 hpi as a baseline control. Virus growth was determined by comparing the presence of vRNA by qRT-PCR or infectious virus by VPA in the supernatant between 0 and 96 h time-points, in addition to immunofluorescent staining of coverslips for the presence of RVFV antigen.

**Statistics**

Two-way ANOVA was performed using GraphPad Prism (version 9.0).

**Reporting summary**

Further information on research design is available in the Nature Portfolio Reporting Summary linked to this article.

## Data availability

All data needed to evaluate the conclusions in this manuscript are present in the manuscript and/or the Supplementary Materials. The data generated in this study was stored in Microsoft Excel for Microsoft 365 MSO (Version 2306 Build 16.0.16529.20100) and graphed using GraphPad Prism v9. Source data are provided with this paper.

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

## Acknowledgements

We would like to thank Reagan Walker for advice on experimental design and Stacey Barrick for study coordination. We also show our appreciation to the Center for Biological Imaging and the McGowan Center for Regenerative Medicine for histology support. All work was supported by the National Institutes of Health (R01AI150792 to A.L.H.; 75N93019C00062 to J.E.C.). The funders had no role in study design, data collection and analysis, decision to publish, or preparation of the manuscript. Graphics were created using biorender.com per license agreement number: GP25KKM1I5.

## Author contributions

Conceptualization: C.M.M., N.S.C., J.E.C., A.L.H. Data curation: C.M.M., N.S.C., R.M.H., L.B.S., M.M.S., L.S.H., J.E.C., A.L.H. Formal analysis: C.M.M., N.S.C., J.E.C., L.B.S., A.L.H. Funding acquisition: J.E.C., A.L.H. Investigation: C.M.M., N.S.C., R.M.H., L.B.S., M.M.S., L.S.H., J.E.C., A.L.H. Methodology: C.M.M., N.S.C., R.M.H., L.B.S., M.M.S., L.S.H. Project administration: J.E.C., A.L.H. Supervision: J.E.C., A.L.H. Validation: J.E.C., A.L.H. Visualization: C.M.M., J.E.C., A.L.H. Writing – original draft: C.M.M., N.S.C., J.E.C., A.L.H.

## Competing interests

J.E.C. has served as a consultant for Luna Labs USA, Merck Sharp & Dohme Corporation, Emergent Biosolutions, GlaxoSmithKline and BTG International Inc, is a member of the Scientific Advisory Board of Meissa Vaccines, a former member of the Scientific Advisory Board of Gigagen (Grifols) and is founder of IDBiologics. The laboratory of J.E.C. received unrelated sponsored research agreements from AstraZeneca, Takeda, and IDBiologics during the conduct of the study. The remaining authors declare no competing interests.
