## [Peer Review File · Nature Communications]

A highly potent human neutralizing antibody prevents vertical transmission of Rift Valley fever virus in a rat modelReviewers' Comments:

Reviewer #1:

Remarks to the Author:

The paper by McMillen et al. reports the ability of a rift valley fever virus-specific antibody to protect dams and fetal rats from death and unconstrained viral replication. A rigorous control (human mAb) is used, and in vivo experiments include multiple dosing and timing schemas. The work is generally well-described and some limitations articulated. The work is impactful given the need for effective interventions against this pathogen, and the broader prospects of maternal antibody delivery for a variety of diseases that bear especially severe sequelae in the context of early life infection.

For improvement:

Major points:

Stats and subject n values are not clearly described. This point needs to be remedied. For example, since n are different across figure 3, it would be good to list n below each measurement. Same goes for Figure 2B, 5B, 6B, 6C. Additionally, noting the correspondence between dams and fetal tissue samples would be helpful. For example, the infectious virus in viscera of control mAb treatment in 3B appear bimodally distributed. Are all the fetuses with higher levels from the same dam?

More detail on histology is needed. There is no statement that sections were analyzed by a blinded viewer, quantitative analysis, or description of representative images being shown, or description of the number of livers analyzed. One is left to presume this figure has low rigor.

Rat FcRn and human Fc are described as "species incompatible". The meaning of this description needs to be provided.

Detection of the control mAb would inform as to whether clearance of mAb RVFV-268 is unusual or consistent with another human IgG1. It is unclear what clearance rate is expected in non-pregnant animals, making speculation as to impacts of changes in FcRn expression levels (which would arguably generically extend halflife) feel quite tenuous.

Check citation placement. For example on line 452, citation placement suggests the reference reports transgenerational transfer in rats, but the reference is a PK study in (non-pregnant) NHP. Additionally, there are many reasons to suspect that YTE mutations would not improve transfer in rats as their FcRn does not show the same binding properties to YTE as human FcRn. Poorer PK and transfer of such mutants has been established in mice. A recent publication reports results of transfer experiments with halflife extended Abs in primates (Rosenberg et al., mBio, 2023) whose observations may be relevant to this notion.

It may be appropriate to mention other settings in which maternal Ab therapy could primarily protect the fetus/infant, such as HSV. Discussion of the practicality of such an intervention would round out the discussion. What is known about time of exposure vs. time of placental/fetal infection? What are the PK properties of transfer of human IgG1 to placenta and fetal tissue in humans? Some of these factors are known, but they are not discussed.

Minor points.

Opening sentence in the abstract should be revised to avoid suggestion that RVFV only causes severe disease in Africa and the Middle East. Specifying that these regions are where it circulates (rather than leaving more open the possibility that this geography is driven by differences in human susceptibility) seems appropriate.

It would be helpful to note in the abstract or introduction what family of viruses RVFV falls into.

Line 218: is with multiple comparisons correction meant? If so, please state which method was used. If not, please describe as unadjusted or omit at least omit "with multiple comparisons".

Definition of error bars in figures is needed.

Replicate structure is sometimes unclear. For t figure 1C, does n=3 indicate 3 slices/wells? Or data collected from 3 placentas?

In the experiments defining antibody levels in dam and fetal tissues, can contamination of tissue fractions with RVFV-268 during sample prep be excluded?

Lines 292-298 are confusing. The sentence begins with a subject of "dams that... were euthanized at 3dpi". I think everything is described correctly, it just took me quite a while to make sure that I understood. The authors might consider graphing the dam survival and sacrifice timepoint information in Figure 3A underneath the timeline.

The statement that "mAb therapeutics can be safe and regularly used during pregnancy should be supported by a citation" (line 397). It would also benefit from narrowing, as many mAbs are not considered safe during pregnancy.

The acronym "LOD" should be defined in the figure legend. (And the upper bound should probably be given a distinct acronym (like ULOQ, upper limit of quantitation or whatever would be appropriate).

Reviewer #2:

Remarks to the Author:

Fetal infection and pregnancy loss are key pathogenic features of RVFV infection in livestock and likely in humans as well. Neutralizing mAbs are potentially valuable interventions against RVFV but it is important to characterize their activity during pregnancy to assess their utility in preventing adverse fetal outcomes. Here McMillen and colleagues evaluated a highly neutralizing mAb in a rat pregnancy model that exhibits pathology and pregnancy loss after RVFV infection. These experiments are challenging to perform (given the biocontainment and biosecurity requirements for RVFV and the added difficulty of a pregnancy model). Overall these experiments provide useful new information about the use of neutralizing mAbs in the context of RVFV infection in pregnancy. The manuscript is clearly written and conclusions are careful; the figures are very nicely presented. I have only minor suggestions for improvement.

Minor text/figure edits:

Line 43: delete "Saudi"

Line 44: delete "Africa"

Line 49: change "symptoms" to "disease signs"

Line 52: delete "Zoonotic"

Line 58-59: what happened to the other infant? (there are only 2)

Line 69: "prevent disease from human..."

Line 79: delete "in vivo"

Line 109: delete "was produced"

Line 268: should "post delivery" be "post harvest"?

Line 374: placenta samples

Line 455: tissue resident

Line 685: Dr. Crowe is listed as contributing to funding acquisition but the only grant listed is to Dr. Hartman.

Graphs throughout: samples that are not detected should be plotted on the LOD line (since that is the sensitivity of the assay)

We thank you and the reviewers for your careful consideration of our manuscript and for the recommendations outlined below. We are happy to hear that you recognize the impact our findings have on the field. As described below, we have addressed the comments raised by the reviewers and strongly believe that these edits have improved the rigor and clarity of the manuscript, thus making the manuscript ready for publication. Specific responses to reviewer comments are noted below in red:

REVIEWER COMMENTS

Reviewer #1 (Remarks to the Author):

The paper by McMillen et al. reports the ability of a rift valley fever virus-specific antibody to protect dams and fetal rats from death and unconstrained viral replication. A rigorous control (human mAb) is used, and in vivo experiments include multiple dosing and timing schemas. The work is generally well-described and some limitations articulated. The work is impactful given the need for effective interventions against this pathogen, and the broader prospects of maternal antibody delivery for a variety of diseases that bear especially severe sequelae in the context of early life infection.

For improvement:

Major points:

Stats and subject n values are not clearly described. This point needs to be remedied. For example, since n are different across figure 3, it would be good to list n below each measurement. Same goes for Figure 2B, 5B, 6B, 6C. Thank you for pointing out the lack of clarity regarding the statistical information. Statistical analyses and p values have been defined in Figure 1 and Supplemental Figure 1 legends as well as the methods section (Line 225). N values for each tissue or sample type have been added to the bottom of each graph and removed from figure legend.

Additionally, noting the correspondence between dams and fetal tissue samples would be helpful. For example, the infectious virus in viscera of control mAb treatment in 3B appear bimodally distributed. Are all the fetuses with higher levels from the same dam? Thank you for pointing this out. We updated the graphs so that the dams and corresponding fetal tissue are indicated by the same-colored symbol on all relevant graphs.

More detail on histology is needed. There is no statement that sections were analyzed by a blinded viewer, quantitative analysis, or description of representative images being shown, or description of the number of livers analyzed. One is left to presume this figure has low rigor. Thank you for this suggestion. Line 177-178 had already stated that "Microscopic histopathologic examination was performed by a board-certified anatomic pathologist, blinded to treatment groups". On line 192 we added the following statement "Representative images are shown for each group including images from an uninfected and untreated dam as a control." In

the legend of figure 4, we included the n value and the following statement “Panels are representative pictures of a liver from each treatment group from figure 3. Livers from uninfected dams (n=2) not treated with mAb were controls.”

Rat FcRn and human Fc are described as “species incompatible”. The meaning of this description needs to be provided. To define “species incompatible” we added the following statement starting on Line 447 “The binding affinity between the Fc of one species and FcRn of another species varies (herein, species incompatibility), and as a result can impact antibody cycling and serum concentrations.”

Detection of the control mAb would inform as to whether clearance of mAb RVFV-268 is unusual or consistent with another human IgG1. It is unclear what clearance rate is expected in non-pregnant animals, making speculation as to impacts of changes in FcRn expression levels (which would arguably generically extend half-life) feel quite tenuous. Thank you for making this astute point. To emphasize your point that clearance may be dependent on pregnancy status, we added the statement “Further studies directly comparing the clearance rate of mAb RVFV-268 in pregnant and non-pregnant animals are warranted” (Line 454). We believe this extends and complements the statement that precedes. With respect to the question as to whether the clearance rate is mAb RVFV-268 specific, we revised Line 456 to “General features of this antibody, such as, paratope charges and isoelectric point may also contribute to the clearance rate observed in this model”.

Check citation placement. For example on line 452, citation placement suggests the reference reports transgenerational transfer in rats, but the reference is a PK study in (non-pregnant) NHP. Additionally, there are many reasons to suspect that YTE mutations would not improve transfer in rats as their FcRn does not show the same binding properties to YTE as human FcRn. Poorer PK and transfer of such mutants has been established in mice. A recent publication reports results of transfer experiments with half-life extended Abs in primates (Rosenberg et al., mBio, 2023) whose observations may be relevant to this notion. Thank you for catching the error in the citation. The citation has been moved to an appropriate location (Line 460). We also added clarification with regards to how YTE mutations may or may not improve transfer in rats (lines 464-468) and we cited Rosenberg et al as suggested.

It may be appropriate to mention other settings in which maternal Ab therapy could primarily protect the fetus/infant, such as HSV. Discussion of the practicality of such an intervention would round out the discussion. What is known about time of exposure vs. time of placental/fetal infection? What are the PK properties of transfer of human IgG1 to placenta and fetal tissue in humans? Some of these factors are known, but they are not discussed. This is a great suggestion. We included a paragraph describing the use of mAbs to prevent vertical transmission of HSV starting on line 494.

Minor points.

Opening sentence in the abstract should be revised to avoid suggestion that RVFV only causes severe disease in Africa and the Middle East. Specifying that these regions are where it circulates (rather than leaving more open the possibility that this geography is driven by

differences in human susceptibility) seems appropriate. We agree with this statement therefore “Severe” was removed from the text (Line 24).

It would be helpful to note in the abstract or introduction what family of viruses RVFV falls into. We added (family *Phenuiviridae*, genus *phlebovirus*)(Line 44).

Line 218 (now line 226): is with multiple comparisons correction meant? If so, please state which method was used. If not, please describe as unadjusted or omit at least omit “with multiple comparisons”. “with multiple comparisons” was removed.

Definition of error bars in figures is needed. Error bars are defined as standard deviation in Figure 1 and Supplemental Figure 1.

Replicate structure is sometimes unclear. For figure 1C, does n=3 indicate 3 slices/wells? Or data collected from 3 placentas? For experiments performed in figure 1, we included the following additional information for clarification: “For prophylactic studies, three placentas were collected from one dam. Two slices from each placenta (n = 6 slices per group) were distributed to each therapeutic group,” and “Supernatant from each placenta was collected and pooled at 0, 24, 36 and 48 hpi, resulting in three biological replicates per group”

In the experiments defining antibody levels in dam and fetal tissues, can contamination of tissue fractions with RVFV-268 during sample prep be excluded? Yes

Lines 292-298 are confusing. The sentence begins with a subject of “dams that... were euthanized at 3dpi”. I think everything is described correctly, it just took me quite a while to make sure that I understood. The authors might consider graphing the dam survival and sacrifice timepoint information in Figure 3A underneath the timeline. We removed the repetitive statement “received antibodies 2 hours prior to virus inoculation and” to improve clarity. The euthanasia time points (3dpi & 6dpi) are clearly indicated on Figure 3A and Figures 3B&C.

The statement that “mAb therapeutics can be safe and regularly used during pregnancy should be supported by a citation” (line 397). It would also benefit from narrowing, as many mAbs are not considered safe during pregnancy. The word “safe” was removed and the current statement is now supported with a citation (Pham-Huy et al. 2021).

The acronym “LOD” should be defined in the figure legend. (And the upper bound should probably be given a distinct acronym (like ULOQ, upper limit of quantitation or whatever would be appropriate). LOD and ULOQ were defined in the figure legend for figure 1 and the methods (Line 211), then the acronym for LOD was used in the proceeding figure legends.

Reviewer #2 (Remarks to the Author):

Fetal infection and pregnancy loss are key pathogenic features of RVFV infection in livestock and likely in humans as well. Neutralizing mAbs are potentially valuable interventions against

RVFV but it is important to characterize their activity during pregnancy to assess their utility in preventing adverse fetal outcomes. Here McMillen and colleagues evaluated a highly neutralizing mAb in a rat pregnancy model that exhibits pathology and pregnancy loss after RVFV infection. These experiments are challenging to perform (given the biocontainment and biosecurity requirements for RVFV and the added difficulty of a pregnancy model). Overall, these experiments provide useful new information about the use of neutralizing mAbs in the context of RVFV infection in pregnancy. The manuscript is clearly written and conclusions are careful; the figures are very nicely presented. I have only minor suggestions for improvement.

Minor text/figure edits: All recommendations below were addressed in the manuscript unless otherwise noted. Thank you for identifying these errors.

Line 43: delete "Saudi"

Line 44: delete "Africa"

Line 49: change "symptoms" to "disease signs"

Line 52: delete "Zoonotic"

Line 58-59: what happened to the other infant? (there are only 2) **The parents discharged the infant from the hospital against the hospital's recommendations, therefore the outcome of the infant is not known. We believe adding this additional piece of information distracts from the overall point of this statement and choose not to expand further in the text.**

Line 69: "prevent disease from human..."

Line 79: delete "in vivo"

Line 109: delete "was produced"

Line 268: should "post delivery" be "post harvest"? **We added post-delivery "of the antibody"**

Line 374: placenta samples

Line 455: tissue resident

Line 685: Dr. Crowe is listed as contributing to funding acquisition but the only grant listed is to Dr. Hartman. **Dr. Crowe's grant information is now listed (Line 680).**

Graphs throughout: samples that are not detected should be plotted on the LOD line (since that is the sensitivity of the assay) **Thank you for this comment. The LOD is the limit of detection, i.e. the lowest detectable amount. Since nothing amplified, then the LOD does not correspond with the sample quantity. For example, the lowest number of plaques we can detect is 25pfu/mL but if we do not have any plaques, then we think the data point should be placed below the LOD, and thus we place it at ND to distinguish between the two. We clarified this in the methods section: For plaque assay and q-RT-PCR data, ND = none detected and is defined as anything below the LOD (determined based on lowest number of countable plaques or lowest standard curve that amplifies, respectively).**

Reviewers' Comments:

Reviewer #1:

Remarks to the Author:

The responses to review are appropriate and appreciated, but a marked up copy of the manuscript did not appear to be available, and line numbering on the provided draft did not correspond to the version that could be viewed. Therefore, I could not easily confirm the accuracy of the responses and changes to the manuscript in context.